# Staff’s Knowledge and Self-Confidence in Difficult Communication: Evaluation of a Short Experiential-Based Training Program

**DOI:** 10.3390/nursrep15020060

**Published:** 2025-02-08

**Authors:** Silvia Gonella, Ludovica Brofferio, Luigi Stella, Daniele Sciarrotta, Paola Di Giulio, Valerio Dimonte

**Affiliations:** 1Department of Public Health and Pediatrics, University of Torino, 10126 Turin, Italy; 2Fondazione Assistenza e Ricerca Oncologica (F.A.R.O.), 10126 Turin, Italy; 3Assi.I.S.Te Company, 10123 Turin, Italy; 4City of Health and Science University Hospital of Turin, 10126 Turin, Italy

**Keywords:** communication skills training, difficult communication, experiential learning methods, knowledge, nursing home, short-term outcomes, satisfaction, self-confidence, self-preparedness, video

## Abstract

**Background:** Most of the communication training programs developed over the past two decades were monodisciplinary, relied on traditional teaching methods, and targeted the hospital context. **Objectives:** The aim of this study is to assess the impact of a short, interdisciplinary, experiential-based communication program (Teach-to-Communicate program) targeted at nursing home (NH) personnel with regard to short-term, staff-related outcomes. **Methods:** This study is part of a larger quality improvement project based on a pre–post single-arm intervention design. We focused on a 6 h residential program involving 30 participants with different scopes of practice working in an NH. Traditional and experiential learning methods were used, including lectures, small group discussions, brainstorming sessions, videos on successful and failed communication, role play, and storytelling based on real cases. The primary outcome was the acquired knowledge of communication strategies and protocols (pre- and post-test quiz). Secondary outcomes were self-reported knowledge, preparedness, confidence, and satisfaction with training (5-point Likert questions). **Results:** A statistically significant improvement in acquired knowledge with a large size effect (0.7, *p* < 0.001) was observed. Self-reported preparedness and confidence ameliorated for all measured communication skills and topics, with the highest effect size registered for self-reported preparedness and confidence in engaging in difficult communication (both 0.7, *p* < 0.001). Participants were highly satisfied with the training, and particularly with the use of video cases (mean 4.6, SD 0.6) and the relevance to clinical practice (mean 4.7, SD 0.5). **Conclusions:** The Teach-to-Communicate program holds promise for improving knowledge and self-confidence regarding difficult communication among NH staff, and highlights the utility of video cases in communication skills training.

## 1. Introduction

Nursing homes (NHs) are one of the main settings of complex communication scenarios such as the communication of diagnosis or illness progression, accommodation of strong emotions, facing unrealistic care expectations or illness unawareness, and eventually managing contrasting care preferences among family carers or between the resident and their family [1].

At some stage of their institutionalization, NH residents are likely to experience a high symptom burden, functional decline, and organ failure [2,3]. It has been estimated that approximately 70% of NH residents have criteria relating to a transition to palliative-oriented care [4] and up to 80% are no longer sufficiently cognitively competent to share their care preferences [5]; thus, family carers often become a key partner in the provision of quality and goal-concordant care [1].

Navigating difficult communication sensitively facilitates therapeutic alliance and the provision of person-centered care that is aligned with a resident’s preferences and values, promotes understanding of the clinical conditions and disease trajectory, and supports a smooth transition towards palliative-oriented care with attention to symptoms burden and family carers’ informational and supportive care needs [1]. Quality communication results in positive attitudes towards healthcare organizations and greater satisfaction with care from both patients and their families [6,7]. Conversely, ineffective communication can lead to unbeneficial, disproportionate, or even harmful care [8]. Interdisciplinary cooperation is key in difficult communication, bringing together different points of view and sharing the emotional burden of decision-making, and finally communicating effectively [6,9].

Despite the widespread consensus that effective communication positively impacts the quality of care [10], in several countries, including Italy, healthcare professionals often receive little or no training related to holding difficult conversations. Moreover, diverse barriers can obstacle effective communication including lack of time; fragmented or poor communication within the team; professionals’ feelings of fear, anxiety, or inadequacy; difficulties in prognostication; limited or inappropriate documentation related to patients’ values, wishes, and care preferences; and poor clarity regarding professionals’ role, which may lead to conflicting or confusing information being shared [11,12]. Entering end-of-life discussions requires professionals to confront their own situation and mortality, and different cultural backgrounds or language barriers could potentially complicate such conversations [13]. However, good relationships with and knowledge of patients and their loved ones, the presence of advance directives or substitute decision-makers, as well as professional experience and communication skills training, including mentorship in the form of watching supervisors, positively influence self-confidence [13,14].

Several communication skills training interventions have been developed over the past two decades; however, most of the programs were monodisciplinary and targeted physicians or medical residents, mainly relied on traditional teaching methods, and were limited to specific theoretical communication models [15,16]. Moreover, interventions were mainly bounded to the acute care setting, and none targeted the NH staff [17].

To improve the quality of communication in NH, a quality improvement project based on a more comprehensive intervention called the Teach-to-Communicate program was developed. This program is composed of an entry-level and an advanced course, adopts both traditional and experiential teaching methods, and is framed within an innovative pedagogical approach that combines the teaching of communication techniques and strategies with the development of relational and self-reflection skills in an interdisciplinary setting [18]. Communicative competence is closely related to self-reflective and self-awareness skills [15]; therefore, experiential methods should be used to complement traditional learning methods to give value to experience and reflective practice. Moreover, an interdisciplinary learning environment has been recognized as key to promoting quality communication, a sense of belonging, and mutual understanding of each specific professional role and scope of practice [19].

The Teach-to-Communicate program aims to achieve both short-term and distal outcomes. The former refer to the immediate or initial changes experienced by participants, which are often related to the knowledge or awareness of the target group. The latter are the final or ultimate changes that take place among participants or the wider community, may occur after years, and are linked to the long-term benefits, impact, and sustainability of the program [20]. In the Teach-to-Communicate program, short-term outcomes relate to participants’ acquired knowledge of communication strategies and protocols, self-reported knowledge, confidence in engaging in difficult communications, and satisfaction with the educational intervention. Distal outcomes refer to the quality of communication reported by family carers, their emotional burden and satisfaction with the care received, and the completion of any advanced care planning documents [18].

Despite the limitations of short-term, staff-related outcomes [21], satisfaction with training, acquired knowledge, and self-confidence represent the starting point of evaluation to improve program design and delivery [22]. Knowledge and self-confidence could positively influence their motivation and sensitivity with regard to difficult conversations and promote the transfer of contents into practice with a gradual cultural change [23,24].

## 2. Teach-to-Communicate Intervention

The Teach-to-Communicate intervention is a two-level educational program: entry and advanced. The entry program is targeted at all NH personnel, including clinical staff, non-clinical staff, and clerks, with the goal of promoting knowledge on how to engage in difficult communication. The entry course involves one 6 h residential program and includes the following: (a) A lecture on clinical, psychosocial, and ethical–deontological issues related to difficult communication. (b) A brainstorming session focused on lived experiences of successful and failed communication. (c) Videos based on a real-life clinical scenario in an NH—a family meeting focused on short-term prognosis in a cachectic resident with advanced dementia—that point out the most frequent communication challenges and communication skills. Both bad and good versions of the videos were shown: the former provided examples of ineffective and poor-quality communication and were presented first to promote reflection and discussion among participants; the latter offered examples of appropriate use of communication strategies. (d) Small group discussions. A multidisciplinary panel of nurses, a psychologist, a palliative care physician, and a bioethicist led the educational intervention [18].

The advanced program is limited to professionals who participate in care planning meetings with residents and/or their family carers and aims to improve relational and communication skills in complex situations. The course is 16 h in length, runs over a three-month period, and offers highly experiential education: (a) a forum theater workshop, i.e., a form of theater where professionals being trained initially take the role of spectators and then are called to intervene and seek solutions by replacing the actors; (b) a simulation workshop on difficult communication scenarios; (c) in-the-field family care conferences followed by feedback from a colleague who acted as an external observer and a guided reflection; (d) a community of practice where professionals have the opportunity to share the problems that emerged during the family care conferences and the strategies they adopted; and (e) online asynchronous self-educational activities with a discussion forum moderated by a communication expert [18].

Both programs were recognized for Continuing Medical Education (CME) credits by the FARO CME provider, according to the Italian system of lifelong learning.

## 3. Objective

The primary aim of this study was to assess the impact of the Teach-to-Communicate entry program on participants’ acquired knowledge of communication strategies and protocols. Secondary aims related to self-reported knowledge, preparedness, and confidence in engaging in difficult communications, and satisfaction with the educational intervention.

## 4. Methods

### 4.1. Study Design

This study is part of a larger quality improvement project [18] based on a pre–post single-arm intervention design and is aimed at improving the quality of communication in NH through context-tailored communication skills training, thus addressing a knowledge–action gap [25]. This study focuses on the evaluation of the “Teach-to-Communicate entry program” on staff-related outcomes immediately after the training.

The Ethics Committee of the University of Torino, Italy (reference number 0675977, approval date 23 December 2021) approved the study.

### 4.2. Context

The Teach-to-Communicate entry program was carried out in November 2023 in one North-West Italian NH that participated on a voluntary basis. This was a 53-bed, privately led, urban NH with a 1:16 nursing personnel-to-resident ratio that is representative of NHs in Italy [26]. An internal reference person provided NH staff with written information about the purpose of the quality improvement project in the month prior to the educational entry course.

### 4.3. Content and Lecture Hours of the “Teach-to-Communicate” Entry Course

The 6 h entry course comprised the following activities: (a) a warming-up opening with presentation of the participants and the quality improvement project (30 min); (b) a 1 h lecture on the meaning of difficult communication associated with brainstorming of failed and successful communication experiences. A special focus was placed on accommodating emotions and care preferences; (c) a 1 h lecture on clinical issues related to difficult communication (e.g., unpredictable disease trajectory, difficulties in prognostication); (d) a 1 h lecture on ethical and deontological issues (e.g., exploring residents’ values and wishes), and the key role of family meetings; (e) a 1 h lecture on communication barriers, communication strategies, techniques, and protocols; (f) a 1 h period dedicated to watching videos highlighting different communication challenges and communication skills abilities, accompanied by discussions held in small groups of around ten participants each; and (g) an evaluation test (30 min).

### 4.4. Sampling and Participants

Although all NH personnel were eligible, only one edition of the entry course was scheduled, and the NH director had to decide which staff members were offered the opportunity to participate in the course while ensuring residents’ care was unaffected. Thirty-four potential participants out of fifty-nine staff members were initially identified to join the entry course. Ultimately, 4 staff members missed the course due to sick leave and 30/59 participated: 18 non-clinical staff members, 4 nurses, 3 physicians, 2 clerks, 1 physiotherapist, 1 psychologist, and 1 occupational therapist. Most of the participants (*n* = 25/30) had at least 10 years of working experience and 13 had at least 5 years of experience working in that specific facility; 57% (*n* = 17) had a permanent employment contract. Only two participants had previously attended communication training courses, none of which were specifically focused on conversations about serious illness (Table 1).

### 4.5. Data Collection

All 30 participants provided written informed consent and completed paper and pencil questionnaires immediately before and immediately after the 6 h of training. Demographic and professional characteristics were also collected.

### 4.6. Model of Evaluation and Outcome Measures

The educational impact of the course was evaluated according to the first two levels of the Kirkpatrick’s model, which is deemed an appropriate model for evaluating educational programs [27].

The first level of this model assesses the participants’ reaction to the training experience; thus, at the end of the entry course, the participants completed a purpose-designed 9-item satisfaction questionnaire to assess organization of the training (4 items), training methods (3 items), and professional relevance (2 items), on a 5-point Likert scale from 1 (Strongly disagree) to 5 (Strongly agree).

The second level of the Kirkpatrick model evaluates the trainee’s learning outcomes. In this step, the participants were assessed immediately before and immediately after the course for:

*Acquired and self-reported knowledge of communication strategies and protocols*: An evaluation test involving a quiz with 30 true or false questions was purpose-designed based on the content of the entry course. The questions covered three thematic areas: (1) setting up and conducting family meetings (9 items); (2) communication barriers and effective communication strategies and techniques (12 items); and (3) accommodating the emotions and care preferences of the resident and/or their family carers (9 items). A correct answer was given a score of 1, while an incorrect response scored 0. Thus, the possible range of total scores was 0–30 and participants passed the test if they provided at least 18/30 correct answers. As these questions assessed knowledge across three topics rather than measuring a single construct, they were considered to be an index not a scale, and therefore internal consistency on the acquired knowledge measure is not reported.

Participants also self-reported acquired knowledge of communication protocols and strategies on a 5-point Likert scale from 1 (Strongly disagree) to 5 (Strongly agree).

*Self-reported preparedness and confidence in engaging in difficult communication*: A 9-item scale was purpose-designed to assess preparedness and confidence in difficult communication. Participants described how prepared they felt about “dealing with difficult communication”, “exploring residents’ and their family carers’ care values and wishes, their awareness of the clinical situation, their desire to know about the disease, potential trajectory, and possible treatments”, and “documenting family meetings” (Mc Donald Omega = 0.868). Moreover, they reported how confident they felt about “engaging in difficult communication”, “accommodating emotions”, “successfully managing conflicts with residents and/or their family carers when care values differ”, and “closing a difficult communication” (McDonald’s Omega = 0.818). Each item was scored on a scale from 0 (not prepared/not confident) to 5 (highly prepared/highly confident). The McDonald’s Omega for the overall scale was 0.913, representing excellent internal consistency [28].

### 4.7. Data Analysis

Descriptive and inferential statistics were calculated. Categorical variables are expressed as frequencies and percentages; continuous variables are expressed as means with standard deviation (SD). The Shapiro–Wilk test was used to verify the normality of the distribution. The Wilcoxon signed-rank test was used to assess changes in gained knowledge and self-reported preparedness and confidence as continuous measures, as well as changes in the proportion of correct answers for each thematic area of the knowledge evaluation test. The product–moment correlation ($r=z/\sqrt{N}$) estimated the intervention effect size (|r| < 0.10 very small; 0.10 ≤ |r| < 0.30 small; 0.30 ≤ |r| < 0.50 medium; |r| ≥ 0.50 large) [29].

The total score for gained knowledge was categorized into four classes (<18, 18–21, 22–25, and ≥26) and the Wilcoxon signed-rank test was used to assess changes from baseline to post-intervention based on the quiz performance. Only the participants with both pre- and post-training course data were considered in the inferential analyses. A *p*-value < 0.05 was considered statistically significant. IBM SPSS version 28 was used for all statistical analyses.

## 5. Results

### 5.1. Acquired and Self-Reported Knowledge of Communication Strategies and Protocols

Twenty-eight participants completed the pre-intervention test and 30 the post-intervention test. Six participants failed the pre-intervention test and one failed the post-intervention test. A significant difference in test success was observed between pre- and post-intervention results (*p* = 0.059).

The mean post-test score was significantly higher than the pre-test score (23.5 (SD 2.9) vs. 21.3 (SD 4.5), *p* < 0.001, effect size 0.7). Overall, performance improved for 19/30 (63.3%) participants, worsened for 4/30, and did not change for 5/30 (*p* = 0.001) (Table 2).

In the pre-test, a nurse had the highest score (29/30) and two non-clinical staff members the lowest scores (both 13/30). In the post-test, three nurses, a psychologist, and a clerk achieved the best scores (all 27/30).

In the post-intervention test, the number of correct answers increased in all the thematic areas, though there were some differences. With regard to thematic area 1 “setting up and conducting family meetings”, participants provided an average of 7.6/9 (1.2) correct answers in the pre-test compared to 8.1/9 (1.2) in the post-test (*p* = 0.053). For thematic area 2 “communication barriers and effective communication strategies and techniques”, an average of 8/12 (2.0) vs. 9.4/12 (1.4) correct answers were obtained for the pre- and post-test, respectively (*p* = 0.002). No significant change emerged in the thematic area 3 “accommodating emotions and care preferences” (an average of 5.5/9 (1.7) vs. 6/9 (1.4) correct answers, *p* = 0.102).

Two items associated with thematic area 2 registered the highest increase in the proportion of correct answers: “communication skills can be acquired as well as technical skills” (15/28 correct true answers in the pre-test vs. 24/30 in the post-test) and “a direct communication style with no gradualness in providing information is recommended to avoid misunderstanding and false hope” (15/28 correct false answers in the pre-test vs. 26/30 in the post-test) (Appendix A).

Correct answers related to the appropriate use of silence in difficult conversations (area 2, 16 vs. 17 correct answers) and how to encourage positive expectations to set up care goals consciously (area 3, 5 vs. 9 correct answers) only slightly increased (Appendix A).

Among the participants who completed both the pre- and post-test and perceived acquired knowledge (*n* = 25/28), 13/25 objectively performed better in the post-test by moving from a score class to the next one(s). These participants included eight non-clinical staff members, two physicians, a nurse, and the occupational therapist. The score class did not change for 12/25 participants (five non-clinical staff members, 2 physicians, 2 nurses, 1 clerk, and the physiotherapist). Among the three participants reporting no acquired knowledge, the proportion of correct answers increased for one nurse and one non-clinical staff member and did not change for another nurse.

There was no significant difference in acquired knowledge between participants who perceived that they had gained knowledge (*n* = 25, mean pre-post difference of 2.3/30) compared to those reporting negative or neutral perceptions (*n* = 3, mean pre-post difference of 2.2/30).

### 5.2. Self-Reported Preparedness and Confidence in Engaging in Difficult Conversations

Both self-reported preparedness (*p* = 0.003; effect size 0.6) and self-reported confidence (*p* < 0.001, effect size 0.7) improved after the intervention. Self-reported preparedness and confidence ameliorated for all measured topics and the highest effect size was registered for self-reported preparedness and confidence in engaging in difficult communication (both 0.7, *p* < 0.001); significant improvements were also observed with regard to exploring residents’ and/or family carers’ awareness of the clinical situation (*p* = 0.006; effect size 0.5), their desire to know about the disease, the potential trajectory, and possible treatments (*p* = 0.042; effect size 0.4), and closing a difficult communication (*p* = 0.046; effect size 0.4) (Table 3).

### 5.3. Satisfaction with the Intervention

Overall, the participants were highly satisfied with the training (mean 4.6; SD 0.6). They judged the educational material relevant and perceived the training content to be in line with the program (both mean 4.7; SD 0.5). Most of the participants strongly agreed with the usefulness of the training methods in improving communication skills, particularly the video cases (mean 4.6; SD 0.6). All participants reported that they would use the acquired knowledge in the field and believed that knowledge transfer could potentially improve their practice (Appendix A).

## 6. Discussion

This study aimed to assess the impact of a short, interdisciplinary, experiential-based training program targeted at NH personnel on their knowledge of communication strategies and protocols, self-reported preparedness, and confidence in engaging in difficult communication, and satisfaction with the educational intervention.

Our findings showed the potential efficacy of this experiential training program in improving NH staff’s knowledge on effectively engaging in complex communication scenarios, with a large effect size and more than half of the participants showing better performance.

Experiential learning in small groups is recognized to be more effective for improving knowledge compared to lectures. The ideal group size should be between five and eight people, with six being considered optimal. However, beyond the number of participants, other pedagogical key elements contribute to successful learning, including active participation and purposeful activities [30]. Although our groups were slightly larger compared to the literature recommendations, with around ten participants each, the educational goals of the intervention were carefully explained and agreed on, the room was appropriately set up to ensure multiple and active interactions, and a facilitator guided all educational activities.

Objective, test-based knowledge increased regardless of the subjective perception of improvement. This indicates that participants grasped the fundamentals of quality communication in complex situations and that self-assessment could be uncorrelated with actual performance, probably because even self-assessment is a skill that has to be acquired [31].

As expected, the greatest knowledge improvement was registered for the topics addressed in the videos (e.g., the importance of a gradual communication style) likely because looking at real-based clinical encounters that could be directly transferred to the participants’ daily practice improved memorization [32]. Instead, the topics covered only in the lectures such as specific techniques and strategies for effective communication (e.g., use of silence, “I would like technique”, de-escalation techniques to solve conflicts) and how accommodating emotions and care preferences of the resident and/or their family carers remained hostile even after the intervention. Furthermore, several items related to factors that may facilitate (e.g., managing denial, encouraging positive expectations, family conferences) or hinder (e.g., acknowledging professionals’ fear of not being able to accommodate the interlocutor’s emotions) shared goals of care discussions showed poor or no improvement. We can postulate that more interactive methods such as role-play or storytelling would benefit the internalization of such content. These results suggest that an inductive approach anchored in clinical practice can maximize learning and provide guidance for restructuring the educational intervention. Future editions of the entry course may start with the video-case session, continue with small group discussions, and systematize the educational content in final lectures.

The Teach-to-Communicate intervention represented the first educational opportunity related to the topic of difficult communication for most of the participants and offered a diverse range of theoretical content and insights which could be directly transferred to daily practice. This made participants feel more self-prepared and confident in engaging in complex communication, and could be promising, as it relates to cross-cutting skills for all NH personnel, such as addressing emotions, exploring awareness, and knowing how to effectively conclude a communication. Moreover, some authors sustained that a greater sense of confidence and positive attitudes toward learning favor changes in daily practice [15].

Our participants were satisfied with the training organization and the educational methods. The number of participants (*n* = 30) and the 6 h course duration could be considered a trade-off between the literature recommendations and the need to guarantee the continuity of care in the facility [33]. The experts advise an ideal number of 16–24 learners, especially when experience-based educational methods (i.e., role-play, small group discussion) are adopted [34], and a total training duration of at least 8 h [35]. However, our entry course needed to reach as many personnel as possible simultaneously to provide an essential foundation for effectively managing difficult communication; its shorter duration facilitated participation. Moreover, this entry course is part of a larger program that develops over 6 months, with an overall duration of 22 h [18]. This will allow participants to consolidate knowledge over time and transfer the acquired learning into daily practice.

The Teach-to-Communicate entry program targets the full spectrum of NH staff, smoothing education disparities associated with accessing training. The program dismantles rigid NH hierarchies and aims to inspire a culture change in which staff members can share education opportunities and jointly participate in improvement projects. Education initiatives should facilitate knowledge improvement while reducing education inequities; this can also facilitate understanding of roles and responsibilities in the care team and, ultimately, improve care outcomes [13]. Therefore, the interprofessional learning environment has been confirmed as an essential element for future iterations of the program. Another important consideration is that having more staff trained, and hopefully the entire staff of a facility, can more effectively promote a culture of timely and sensitive engagement in difficult communication. Therefore, future editions will involve planning enough training days to include all the NH staff. This will require preliminary encounters with the NH leadership to ensure facilitation of training delivery. Indeed, leadership committed to practice improvement is key to successfully implementing interventions [36].

Consistent with a previous study [15], our participants appreciated active and experience-based training methods, which appear more effective than traditional strategies in promoting knowledge and skills [8,15]. Also, compared to other experiential methods such as role-play and case discussion [37], the literature recognizes the strong inclusive nature of video cases [38]. First, adults mainly engage in visual learning and prefer to receive information through images; moreover, video cases are time- and space-independent, allowing participants to review and revise content over time according to their learning needs. Finally, video cases activate emotional reactions by providing reality-based scenarios in which the viewers can mirror themselves or consider from a distance [38]. Video cases can help to summarize and fix the main key take-home messages and can be both a didactic method and an aid used to complement other methods [37,38]. In our study, the video cases scored the highest for perceived usefulness, likely because they provided the opportunity to transfer what the learners were taught during the lecture to a real scenario. A previous systematic review suggested that the use of a conversation-analytic approach via audio or video taping could be important for evidence-based and reflective practice because it makes communication practice explicit [39]. A video-taped, real-life communication allows participants to analyze not just what is said, but how it is said, including body language. Furthermore, relevant and realistic video cases can demonstrate the applicability and transferability of educational content into practice. Our entry course used both bad and good communication examples consistent with previous experiences, which demonstrated how video cases are likely to foster communication skills, especially when bad examples are employed, if associated with feedback [40]. The reflective practice was indeed a main pillar of the training program as well as the multidisciplinary collaboration. Our participants were mainly non-clinical staff members and nurses who seemed to have gained awareness of their potential contribution to a team-based approach to complex communication scenarios and of the importance of collectively discussing challenging real-life cases to reflect and learn from the experience. This is supported by the increased correct response rate after the course (always over 80% and up to 100%) to the knowledge test items which related to a team-based approach to preparing and conducting family meetings.

## 7. Methodological Considerations

There are some elements that may affect the reliability and validity of the findings and need to be considered. First, this study is part of a larger quality improvement project and the evaluation of short-term, staff-related outcomes was not originally planned [18]. As the primary goal of quality improvement projects is to improve institution-specific processes rather than detecting differences, no pilot testing and statistical power calculation took place prior to the evaluation. The limited sample size likely prevented us from achieving a significant difference in terms of the number of learners that passed the test after the educational intervention, although a borderline-significant trend emerged. Moreover, it did not allow sub-group analyses to be performed based on demographic characteristics or scope of practice; such analyses would have been helpful for comparing changes among participants and identifying what could be improved in the future courses in relation to the participants’ profiles. Second, the validity of the purpose-designed outcome measures could be questioned, even though their internal consistency was excellent and self-reported knowledge was paired with an evaluation quiz. Third, we acknowledge that the pre–post single-arm design may limit our ability to draw strong causal inferences about the intervention effectiveness and the involvement of only one privately led NH makes it difficult to generalize the findings to other settings. Finally, as previously mentioned, this entry course was limited to the evaluation of short-term, staff-related cognitive and satisfaction outcomes. However, the Teach-to-Communicate intervention is a two-level program, and the advanced course entails the evaluation of the upper levels of the Kirkpatrick model [22]. In particular, the degree to which participants apply what they learnt will be measured by the quality of communication reported by family carers, while the final impact of the intervention will be evaluated by employing indicators such as family carers’ psycho-emotional burden and their satisfaction with the care received, and the completion of advance care planning documents for the residents.

Future iterations of the Teach-to-Communicate entry program need to involve multiple NHs with diverse geographic, cultural, and organizational characteristics, as well as both public and private facilities. All NH staff should have the opportunity to attend the training sessions, with the double aim of promoting a culture change in communication behaviors and minimizing selection bias. Additionally, the evaluation of long-term, distal outcomes related to residents and their families could be introduced into the entry program. Finally, when enough data to draw reflections on the intervention impact based on participants’ characteristics are available, simultaneous breakout sessions tailored to the unique communication needs of different staff members could be set up. This approach can guarantee an interprofessional learning environment while responding to role-specific needs.

## 8. Conclusions

The Teach-to-Communicate entry program, which is a short, multidisciplinary training program, holds promise for improving NH staff’s knowledge and self-confidence regarding difficult communication. It provides preliminary evidence for the integrated use of traditional and experiential teaching methods and highlights the utility of video cases in communication skills training.

## Figures and Tables

**Table 1 nursrep-15-00060-t001:** Characteristics of the participants.

Characteristics	Participants *n* = 30 N (%)
**Age**, years	
≤25	1 (3.3)
26–35	6 (20.0)
36–50	15 (50.0)
≥50	8 (26.7)
**Female gender**	27 (90.0)
**Profile**	
Non-clinical staff	18 (60.0)
Nurse	4 (13.3)
Physician	3 (10.0)
Other *	5 (16.7)
**Working experience**, years	
<1	2 (6.7)
1–4	3 (10.0)
5–10	13 (43.3)
11–15	5 (16.7)
>15	7 (23.3)
**Working experience in the facility**, years	
<1	8 (26.7)
1–4	9 (30.0)
5–10	7 (23.3)
11–15	3 (10.0)
>15	3 (10.0)

* Clerks (*n* = 2), physiotherapist (*n* = 1), psychologist (*n* = 1), occupational therapist (*n* = 1).

**Table 2 nursrep-15-00060-t002:** Knowledge test on communication strategies and protocols.

Knowledge Test	Pre-Intervention Test N = 28	Post-Intervention Test N = 30	*p* Value
**Test passed** *, *n* (%)	22 (78.6)	29 (96.7)	0.059
**Test score**, mean (SD)	21.3 (4.5)	23.5 (2.9)	**0.001**
<18, *n* (%)	6 (21.4)	1 (3.4)	
18–21, *n* (%)	6 (21.4)	4 (13.3)	
22–25, *n* (%)	13 (46.5)	17 (56.6)	
≥26, *n* (%)	3 (10.7)	8 (26.7)	

* Test passed with at least 18/30 correct answers. SD = standard deviation.

**Table 3 nursrep-15-00060-t003:** Self-reported preparedness and confidence in engaging in difficult communication: pre vs. post educational intervention.

	Strongly Disagree to Disagree N (%)		Neutral N (%)		Agree N (%)		Strongly Agree N (%)		Mean (SD) *		r	*p* Value
	PRE	POST	PRE	POST	PRE	POST	PRE	POST	PRE	POST		
**Self-reported preparedness** ** *I feel prepared in* *…* **									**3.1 (0.8)**	**3.5 (0.9)**	**0.6**	**0.003**
dealing with difficult communication. (pre *n* = 29; post *n* = 30)	11 (37.9)	3 (10.0)	13 (43.3)	11 (36.7)	4 (13.3)	11 (36.7)	1 (3.3)	4 (13.0)	2.7 (0.9)	3.5 (0.8)	0.7	**<0.001**
exploring residents and/or their family carers’ care values and wishes. (pre *n* = 29; post *n* = 30)	4 (13.8)	4 (13.3)	11 (37.9)	9 (30.0)	12 (41.4)	13 (43.3)	2 (6.9)	4 (13.3)	3.4 (0.9)	3.5 (1)	0.1	0.613
exploring residents and/or their family carers’ awareness of the clinical situation. (pre *n* = 29; post *n* = 29)	4 (13.8)	4 (13.8)	11 (36.7)	9 (30.0)	12 (40.0)	13 (43.3)	2 (6.7)	3 (10.0)	3.1 (0.9)	3.6 (1)	0.5	**0.006**
exploring residents and/or their family carers’ desire to know about the disease, potential trajectory, and possible treatments. (pre *n* = 29; post *n* = 29)	5 (17.2)	4 (13.8)	16 (53.3)	7 (23.3)	5 (16.7)	14 (46.7)	2 (6.7)	3 (10.0)	2.9 (0.9)	3.3 (1)	0.4	**0.042**
Accurately documenting family meetings. (pre *n* = 29; post *n* = 30)	5 (17.2)	4 (13.3)	17 (56.7)	13 (43.3)	6 (20.0)	9 (30.0)	1 (3.3)	3 (10.0)	3.0 (0.9)	3.4 (1)	0.3	0.087
**Self-reported confidence** ** *I feel confident in…* **									**3.1 (0.9)**	**3.5 (0.9)**	**0.7**	**<0.001**
engaging in difficult communication. (pre *n* = 29; post *n* = 30)	14 (48.3)	5 (16.7)	8 (26.7)	10 (3.3)	7 (23.3)	10 (33.3)	-	4 (13.3)	2.6 (1)	3.5 (0.9)	0.7	**<0.001**
accommodating residents and/or their family carers’ emotions. (pre *n* = 28; post *n* = 30)	6 (21.4)	1 (3.3)	8 (26.7)	6 (20.0)	10 (33.3)	16 (53.3)	3 (10.0)	3 (10.0)	3.2 (1.2)	3.6 (0.9)	0.4	0.076
successfully managing conflicts with residents and/or their family carers when care values differ. (pre *n* = 29; post *n* = 29)	7 (24.1)	6 (20.7)	10 (33.3)	6 (20.0)	7 (23.3)	16 (53.3)	5 (16.7)	1 (3.3)	3.3 (1.1)	3.4 (0.9)	0.1	0.838
Closing a difficult communication. (pre *n* = 29; post *n* = 30)	6 (20.7)	3 (10.0)	14 (46.7)	13 (43.3)	7 (23.3)	8 (26.7)	2 (6.7)	5 (16.7)	3.2 (0.8)	3.5 (0.9)	0.4	**0.046**
** *Overall, the training improved my…* **												
knowledge in dealing with difficult communication. (*n* = 30) ^$^	-	1 (3.3)	-	2 (6.7)	-	16 (53.3)	-	10 (33.3)	-	4.2 (0.8)	-	-
skills in dealing with difficult communication. (*n* = 30) ^$^	-	2 (6.7)	-	3 (10.0)	-	15 (50.0)	-	9 (30.0)	-	4.1 (0.9)	-	-

* Likert Scale 1–5 (1 = strongly disagree; 5 = strongly agree). ^$^ Item administered only post-intervention. Acronym: SD = standard deviation. *Note.* |r| < 0.10 very small; 0.10 ≤ |r| < 0.30 small; 0.30 ≤ |r| < 0.50 medium; |r| ≥ 0.50 large.

## Data Availability

All data generated or analyzed during this study are included in this published article.

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
