# Peer review of "Staff’s Knowledge and Self-Confidence in Difficult Communication: Evaluation of a Short Experiential-Based Training Program"

_nursrep, 2025, doi:10.3390/nursrep15020060_

Round 1

Reviewer 1 Report

Comments and Suggestions for Authors

Thank you for possibility to review this article. I consider the topic very important. I appreciate the demanding methodology. The authors describe the methodology and the results very carefully in detail. Everything is easy to understand for the reader, but the article is too long. Especially the presentation of the results in 5 tables with a large amount of data and the duplicate presentation of data in the text. I recommend reducing the number of tables. I also recommend reducing the number of keywords. I appreciate the description of the limitations and specific recommendations for practice.

Reviewer 2 Report

Comments and Suggestions for Authors

See comments in attached file.

Reviewer 3 Report

Comments and Suggestions for Authors

This article presents a well-structured and insightful evaluation of the Teach-to-Communicate program, highlighting its potential for enhancing communication skills among nursing home staff. The research design, methodology, and analysis are thorough, but a few areas could be refined to strengthen the study further:

  1. Abstract and Objectives: The abstract is concise and provides key findings; however, including specific numerical outcomes (e.g., improvement percentages) could better demonstrate the intervention's impact.
  2. Introduction: While comprehensive, the introduction could benefit from a clearer delineation of the research gap. Expanding on how this program differs from existing training interventions would enhance its novelty.
  3. Methods:
    • The participant selection process is detailed, but additional clarification on how the sample was representative of the broader nursing home workforce would strengthen generalizability.
    • The decision to focus solely on short-term outcomes is acknowledged; however, incorporating even preliminary data on long-term effects could add depth.
  4. Results:
    • Statistical analysis is robust, but presenting additional visuals, such as graphs comparing pre- and post-intervention results, could improve readability and emphasize key findings.
    • The thematic areas for knowledge improvement are well-described, though further discussion on why some areas showed less improvement would provide valuable context.
  5. Discussion:
    • The discussion effectively interprets the results, but further exploration of barriers to greater knowledge improvement (e.g., time constraints or resistance to new methods) could offer practical insights for future iterations of the program.
    • While the use of video-based training is highlighted as effective, comparing this approach with other experiential methods would enhance the discussion.
  6. Limitations:
    • The limitations are candidly addressed, but including a plan for overcoming these in future research would demonstrate proactive consideration for study advancement.
  7. Formatting and Language: The language is clear and academic, though minor grammatical refinements (e.g., consistent use of tenses) could enhance readability. Formatting issues, such as inconsistent spacing in the tables and references, should be corrected.
  8. Conclusion: The conclusion summarizes findings well but could better emphasize actionable recommendations for integrating the Teach-to-Communicate program into broader nursing education practices.

Incorporating these suggestions could further solidify the article's contribution to advancing communication training in nursing homes.

Reviewer 4 Report

Comments and Suggestions for Authors

Improving nursing home staff’s knowledge and self-confidence in difficult communication: Pilot evaluation of a short experiential-based training program

This interesting study assesses the impact of the “Teach-to-Communicate entry program” on participants’ acquired knowledge of communication strategies and their satisfaction with the course. I think that “difficult conversations” is an important skill that is often overlooked. I like the fact that the course included not only professional healthcare workers, but the full spectrum of the nursing home staff. The authors have made a strong argument for the potential and efficacy of their program in improving this important skill. I think overall the study is clear, well, presented and fulfills its objectives. I have made a few comments below, and as I read though also picked up on a few typos and grammar issues. I hope that the authors find my comments constructive and positive. Congratulations on your work.

Comments

With regard to the sample size, are 30 participants enough to provide statistical power? Did you make any calculataions regarding the required sample size (G*Power)? If so, please include this in the text. If not, please expand the limitations section to mention this.

I don’t like the term “non-professional” to refer to those other than medical professionals (nurses, doctors,…), because they are profesionals. How about using “Clinical Staff” and “Non-clinical Staff”?

I was interested in how the participants demographic characteristics (age, gender, years of experience, etc.) might impact their scores. Could you make any meaningful observations regarding those questions? This information might help us to know who is in greater need of such training or who might strongly benefit from it. This might be beyond the scope of your study, but I think it would be good to include this information in your results and discussion.

Similar to the above point, I think it would be good to discuss in the introduction or discussion sections what factors prepare someone to engage in difficult conversations. For example, what makes someone more willing or able to enter an end-of-life discussion? (https://doi.org/10.1016/j.jpainsymman.2020.08.010) Is it experience, training, etc.? What I liked about your program was the variety of participants it engaged and wonder if your study can shed any further light on this broader question. 

A small suggestions, I think the information in Table 2 might have more strong visual impact as a figure (bar graph, pie charts, etc.,) 

Minor edits 

Line 46: Change: “80% is no longer” to “80% are no longer”

Line 55: Change “Instead” to “Conversely” or “On the other hand”

Line 61: Change: “holding difficult communication” to “holding difficult conversations”

Lines 73-74: Change “This intervention is constitutes of an entry and an advanced course” to “This intervention is composed of an entry-level and an advanced course”

Line 79: Change “methods should complement” to “methods should be used  to complement”

Line 101-102: Change “program -entry and advanced-” to “program: entry and advanced.”

Line 107: Remove the question mark ?

Line 123: Change: “colleague’s feedback that acted” to “colleague’s feedback who acted”

Line 140 Change “yields” to “yielded”

Line 161: Change “statements” to “questions”

Line 173: Change “a 9-item” to “A 9-item” 

Line 183 For clarity, change “– 5” to “to 5”

Lines 205 - 207. For clarity, I would revise as follows “In all, 30 NH staff members participated to the entry program: 18 non-professionals, 4 nurses, 3 physicians, 2 clerks, 1 physiotherapist, 1 psychologist, and 1 occupational therapist.” 

Line 210: Change “and none was” to “none of which were”

Line 202: Change “was employed” to “used for all statistical analyses.”

Lines 223-224. Change “scored the best” and “scored the worst” to “had the highest score” and lowest scores”.

Line 250-255. Please be consistent with writing numbers as either words (two physicians) and/or numerals (8 non-professionals). 

Line 266: Change “improvement” to “improvements”

Line 268: I would change “communication” to “conversation”

Line 304: change “prompts” to “indicates”

Line 306: Change “which should be” to “that has to be”

Line 322: Change “feeling” to “feel”

Line 334-335: I’d revise as follows: “…as many personnel as possible simultaneously to provide the essential foundation for effectively managing difficult communication”

Line 335: It’s not clear what is meant by “its residential nature” 

Line 340: Change “Consistently” to “Consistent”

Line 34:1 change “methods that” to “methods, which”

Line 342: change “the higher” to “highest”

Line 353: Change “feedbacks” to “feedback”

Round 2

Reviewer 2 Report

Comments and Suggestions for Authors

Author Response

See file attached.

Round 3

Reviewer 2 Report

Comments and Suggestions for Authors

Thanks to the authors who answered the questions I had after their first revision. Only one comment remains: Delete the word "proximal" from the manuscript, as this term is used without further explanation in this study. The text is sufficiently/more clear by using 'short-term outcomes'.

Author Response

Thanks to the authors who answered the questions I had after their first revision. Only one comment remains: Delete the word "proximal" from the manuscript, as this term is used without further explanation in this study. The text is sufficiently/more clear by using 'short-term outcomes'.

We thank the Reviewer 2 for this further round of evaluation. Her/his constructive feedback has been precious across the round of revisions and really helpful to improve the quality of our paper. We have deleted the word "proximal" throughout the entire manuscript.